# The Impact of TRAIL on the Immunological Milieu during the Early Stage of Abdominal Sepsis

**DOI:** 10.3390/cancers15061773

**Published:** 2023-03-15

**Authors:** Ann-Kathrin Berg, Elisabeth M. Hahn, Fiona Speichinger-Hillenberg, Annemaria Silvana Grube, Nina A. Hering, Ani K. Stoyanova, Katharina Beyer

**Affiliations:** Department of General and Visceral Surgery, Charité—Universitätsmedizin Berlin, Corporate Member of Freie Universität Berlin and Humboldt-Universität zu Berlin, 10117 Berlin, Germanyani.stoyanova@charite.de (A.K.S.)

**Keywords:** peritonitis, sepsis, CASP, TRAIL, neutrophils, apoptosis

## Abstract

**Simple Summary:**

Sepsis is the leading cause of morbidity and mortality worldwide. It reflects a deficiency in the interplay between the innate and adaptive immune response and leads to an injury of the body’s own tissue with dramatic and long-lasting consequences for the patient. However, understanding the immunological background of peritonitis and abdominal sepsis together with its molecular pathomechanism still remains elusive, leading to limited therapeutic options. The TNF-related apoptosis-inducing ligand (TRAIL) has already been shown to induce neutrophil apoptosis and enhance survival in a murine sepsis model. In the present work, we investigated how neutrophil granulocytes regulate their sensitivity to TRAIL-induced apoptosis during the course of sepsis.

**Abstract:**

Despite intensive scientific efforts, the therapy of peritonitis is presently limited to symptomatic measures, including infectious source control and broad-spectrum antibiotics. Promising therapeutic approaches to reduce morbidity and mortality are still missing. Within the early phase of abdominal sepsis, apoptosis of neutrophil granulocytes is inhibited, which is linked to tissue damage and septic shock. TNF-related apoptosis-inducing ligand (TRAIL) is a promising agent to stimulate neutrophil apoptosis. However, the underlying mechanisms have not been elucidated so far. The objective of the present study was to characterize the molecular mechanisms of TRAIL-stimulated apoptosis in early abdominal sepsis. Therefore, the murine sepsis model Colon ascendens stent peritonitis (CASP) was applied in wild type (WT) and TRAIL knock-out (TRAIL–/–) C57/BL6j mice. Neutrophil granulocytes were isolated from spleen, blood, bone marrow, and peritoneal lavage using magnetic-activated cell sorting. Neutrophil maturation was analyzed by light microscopy, and apoptotic neutrophils were quantified by fluorescence-activated cell sorting (FACS). Western blot and FACS were used to investigate expression changes in apoptotic proteins and TRAIL receptors. The impact of TRAIL-induced apoptosis was studied in vitro. In septic mice (CASP 6 h), the number of neutrophils in the BM was reduced but increased in the blood and peritoneal lavage. This was paralleled by an increased maturation of neutrophils from rod-shaped to segmented neutrophils (right shift). In vitro, extrinsic TRAIL stimulation did not alter the apoptosis level of naïve neutrophils but stimulated apoptosis in neutrophils derived from septic WT and TRAIL–/– mice. Neutrophils of the bone marrow and spleen showed enhanced protein expression of anti-apoptotic Flip, c-IAP1, and McL-1 and reduced expression levels of pro-apoptotic Bax in neutrophils, which might correlate with apoptosis inhibition in these cells. CASP increased the expression of intrinsic TRAIL in neutrophils derived from the bone marrow and spleen. This might be explained by an increased expression of the TRAIL receptors DR5, DcR1, and DcR2 on neutrophils in sepsis. No differences were observed between septic or naïve WT and TRAIL–/– mice. In conclusion, the present study shows that neutrophil granulocytes are sensitive to TRAIL-stimulated apoptosis in the early stage of abdominal sepsis, emphasizing the promising role of TRAIL as a therapeutic agent.

## 1. Introduction

Despite intensive research leading to advanced diagnostics, sepsis accounts for one out of five deaths worldwide [1]. Sepsis is a complex, life-threatening organ dysfunction caused by a defect in the interplay between the innate and adaptive immune system, resulting in a complex cascade of inflammatory and counter-inflammatory responses [2]. In the early stage of sepsis, the host reacts to extracellular bacteria, parasites, fungi, or viruses by producing proinflammatory cytokines such as interleukin-6, interleukin-1, and tumor necrosis factor-alpha (TNF-α) [2]. This initial response is followed by a predominance of anti-inflammatory cytokines such as IL-10, increased lymphocyte apoptosis, and a shift from T-helper 1 to T-helper 2 cells [3]. Studies on this biphasic character of sepsis showed that the coexistence of pro- and anti-inflammatory cytokine production results in a mixed antagonistic response syndrome [4].

TNF-related apoptosis-inducing ligand (TRAIL) is known to have a pleiotropic effect during the course of sepsis; it induces apoptosis in transformed cells but also influences inflammatory responses [5]. Previous studies on a murine model of peritonitis demonstrated that intravenous TRAIL administration enhances the survival rate in the early stage of abdominal sepsis by increasing the migration of effector cells into the peritoneal cavity, thereby improving the elimination of bacteria [6]. In the well-established mouse model of colon ascendens stent peritonitis (CASP), peritonitis is induced by the implantation of a transmural stent. This method is considered a standardized model for polymicrobial sepsis, which causes a primary antagonistic response syndrome [7].

Administered in CASP, TRAIL increases the fraction of apoptotic neutrophils and reduces tissue damage [8]. Neutrophils significantly affect acute inflammation as they are rapidly activated in the early stage of sepsis. Once activated, these defense cells follow a hierarchy of chemotactic molecules to reach the site of inflammation [9]. Therefore, neutrophils are the first line of immune defense, using various mechanisms such as phagocytosis, reactive oxygen species degranulation, and NETosis [10] to eliminate pathogens. However, in sepsis, neutrophil apoptosis is inhibited, and the life span of these cells is prolonged, leading to massive tissue damage, which becomes clinically apparent in the form of septic shock. Therefore, the role of TRAIL and its interaction with neutrophil granulocytes in the early stage of sepsis is intensively discussed [11]. In a former study, we showed that TRAIL treatment reduced sepsis-induced organ injury by reducing the life span of neutrophils [8]. However, so far, it has yet to be entirely understood how TRAIL affects neutrophils of the different peripheral lymphoid organs. The present study addresses these questions and investigates the impact of TRAIL on neutrophils in early abdominal sepsis using the CASP mouse model.

## 2. Materials and Methods

### 2.1. Mice

8- to 12-week-old C57BL/6j mice weighing 20–25 g were purchased from Charles River. TRAIL–/– C57BL/6j were obtained from Amgen (Seattle, WA, USA). The knockout was validated by polymerase chain reaction (PCR). Mice were allowed to adapt for two weeks and received water and food ad libitum. Adequate animal housing was provided by a 12:12 h circadian cycle and constant temperature. Mice were kept in groups of five animals. All animal experiments were approved by the regional authority (Landesamt fuer Gesundheit und Soziales Berlin; G0349/17) and were performed in compliance with the European Union guidelines 2010/63/EU.

### 2.2. Colon Ascendens Stent Peritonitis

Colon ascendens stent peritonitis (CASP) was induced as previously described [12]. Briefly, mice were intraperitoneally anesthetized with ketamine (65 mg/kg body weight), xylazine (13 mg/kg body weight), and acepromazine (2 mg/kg body weight). The abdominal wall was opened by a 1 cm midline incision under sterile conditions. The colon ascendens was exposed, and a venous catheter (16-gauge, B. Braun Melsungen AG, Melsungen, Germany) was inserted into the antimesenteric wall, enabling leakage of intestinal contents into the peritoneal cavity. It was fixed with two sutures (7/0 Ethilon thread; Ethicon, Norderstedt, Germany). To ensure the proper intraluminal positioning of the stent, the stool was milked from the caecum towards the colon ascendens until a small amount appeared. After injecting 0.5 mL of sterile saline solution into the peritoneal cavity for resuscitation, the abdomen was closed using 4/0 vicryl stitches (Ethicon). Mice received buprenorphine (0.1 mg/kg body weight) for analgesia. The same surgeon performed all CASP surgeries.

### 2.3. Isolation and Selection of Neutrophils by Using Magnetic-Activated Cell Sorting

Mice were sacrificed 1, 3, 6, and 12 h after inducing CASP by cervical dislocation under deep anesthesia. Spleen, blood, bone marrow, and peritoneal lavage were collected, and cells were isolated from each organ by passing through a 100 μm nylon mesh (BD Falcon^TM^ cell strainer, BD Biosciences, San Jose, CA, USA). The cells were kept on ice and washed three times with 0.5% bovine serum albumin (BSA) (Biochrom Ag, Berlin, Germany) in phosphate-buffered saline (PBS). Neutrophils were isolated by negative selection using a magnetic-activated cell sorting (MACS) kit (Miltenyi Biotech, Bergisch Gladbach, Germany) according to the manufacturer’s protocol. Cells were stained with antibodies according to the manufacturer’s instructions and passed through a column. Purity was validated by flow cytometric analysis (FACS) using a FITC-conjugated anti-Ly6G antibody (Exibo, Vestec u Prahy, Czech Republic). For protein expression analyses in neutrophils, a positive cell sort was performed 6 h after inducing CASP. Cells were isolated from the spleen and bone marrow, blocked with an FcR Blocking reagent (Miltenyi), and stained with PECy7-conjugated anti-Ly6G (RB6-8C5, Tonbo bioscience) and BV510-conjugated anti-CD45 (30-F11, Biolegend) antibodies. Neutrophils were identified as CD45+, DUMP^−^, CD172a+, CD11b+, F4/80^−^ and Ly6G+ by flow cytometry. The cell count was determined using a CASY cell counter (Roche, Mannheim, Germany). Cell sorting and flow cytometry experiments were performed at the Benjamin Franklin Cytometry Facility- University Medical Center Berlin, Germany. For in vitro experiments, neutrophils were subcultured in RPMI-1640 medium (Life Technologies, Darmstadt, Germany) containing 10% fetal bovine serum and 1% penicillin-streptomycin (Thermo Fisher Scientific) at 37 °C in a humidified atmosphere of 5% CO_2_.

### 2.4. In Vitro Treatment with Recombinant TRAIL

Recombinant soluble mouse TRAIL (purity 95%, <1.0 EU per 1 g, Biomol GmbH, Hamburg, Germany) was dissolved in sterile RPMI-1640 medium (Life Technologies, Darmstadt, Germany). Neutrophils were cultured with different concentrations of TRAIL (0 ng, 100 ng, and 1000 ng per 1 × 10^5^ neutrophils) for 1 h, 3 h, 6 h, and 12 h. Cells were harvested at the respective time points, washed two times with PBS, and stained for the following analysis. As a positive control, apoptosis was induced by staurosporine.

### 2.5. Annexin/Propidium Iodide Assay Using Flow Cytometric Analysis (FACS)

The FITC annexin/propidium iodide apoptosis assay (BD Biosciences) was performed to quantify apoptotic neutrophils. The cells were washed once with PBS. To guarantee that only neutrophils were counted, cells were co-stained with PE-Cy7-conjugated Ly6G antibody (RB6-8C5, Tonbo Biosciences, San Diego, CA, USA). After incubation for 30 min at 4 °C in the dark, the suspension was washed once and resuspended in FITC Annexin V Binding Buffer. After adding the FITC annexin V and propidium iodide solution, cells were incubated for 15 min at room temperature in the dark. The reaction was stopped by adding a binding buffer. Apoptotic cells were quantified using a BD FACSCanto (BD Bioscience) and FlowJo software (Treestar Inc., Ashland, IN, USA).

### 2.6. Immuncy to Chemistry

A Pappenheim staining was performed on blood, bone marrow, and peritoneal fluid harvested 1 h, 3 h, 6 h, and 12 h after CASP surgery. Therefore, a smear slide was made, and staining was carried out using the Hemacolor kit (Merck, Darmstadt, Germany) according to the manufacturer’s instructions. The number and morphology of neutrophil granulocytes were determined in four high-power fields (HPFs, 400× magnification) using a light microscope (Axioskop 2, Zeiss, Jena, Germany). Rod and segmented neutrophils were counted. The mean ± SEM was calculated.

### 2.7. Fluorescent Staining and Flow Cytometry

The TRAIL receptors DcR1, DcR2, and DR5, as well as TRAIL, were analyzed on neutrophils in naïve and septic mice by FACS. Surface staining was performed using antibodies against DcR1 (REA759, Milteny Biotech), DcR2 (mDcR2-1, Biolegend, San Diego, CA, USA), DR5 (REA1233, Milteny Biotech), TRAIL (CD 253) (N2B2, Biolegend), CD45 (30-F11, BD Bioscience), CD172 (P84, eBioscience), CD11b (M1/70, Biolegend), F4/80 (BM8, Biolegend), Ly6G (1A8, Biolegend), and DUMP (live/dead, CD3 (17A2, Biolegend), CD19^+^ (6D5, Biolegend) and CD49b (HMalpha2, BD Bioscience)). Cells were analyzed on a FACS BD LSR Fortessa X-20 (BD Bioscience). Neutrophils were identified as CD45^+^, DUMP^−^, CD172a^+^, CD11b^+^, F4/80^−^, and Ly6G^+^:

### 2.8. Western Blot

Protein expression was analyzed by Western Bot. Neutrophils derived from spleen and bone marrow of naïve and septic mice were lysed to extract the proteins. Therefore, lysis buffer containing 10 mM Tris (pH 7.5), 150 mM NaCl, 0.5% Triton X-100, 0.1% SDS, and a complete protease inhibitor mixture (Roche) was used. Samples of 25 μg of protein was separated by polyacrylamide gel electrophorese and blotted onto PVDF membranes (Bio-Rad, Hercules, CA, USA) using the Trans-Blot Turbo Transfer System (Bio-Rad). The following antibodies were applied: anti-BAX (1:1000), anti-Bcl-xL (1:1000), anti-Bcl-2 (1:750), anti-FLIP (1:1000), anti-Mcl-1 (1:750), and anti-Survivin (1:750), (all from Cell Signaling Technology, Cambridge, UK); and anti-CIAP (1:2000, Abcam, Cambridge, UK). For signal detection, peroxidase-conjugated anti-rabbit or anti-mouse secondary antibodies (Sigma-Aldrich, St. Louis, MO, USA) and a chemiluminescent substrate system (SuperSignal West Pico PLUS, Thermo Fisher Scientific, Waltham, MA, USA) were used. Densitometric quantification was carried out using Image J. β-actin served as an internal loading control. For comparison, protein levels of WT were set as 100%.

### 2.9. Statistical Methods

FACS data were analyzed with FlowJo software v10 (BD, San Jose USA). The two-way analysis of variance was used to compare continuous variables. For 2-group comparisons, an unpaired *t*-test was performed. GraphPad Prism 6 software (GraphPad Prism, edition 8.4.3, San Diego, CA, USA) was used for all calculations. Statistical significance was considered for the following *p*-values: * *p* ≤ 0.05 ** *p* ≤ 0.01 *** *p* ≤ 0.001. Data are expressed as the standard error of the mean ± SEM.

## 3. Results

### 3.1. Sepsis Decreases the Number of Neutrophils in the Bone Marrow and Increases the Number of Neutrophils in the Blood and the Peritoneum

Initially, we tested the impact of abdominal sepsis on the neutrophil ratio in the CASP model. Therefore, neutrophils from the bone marrow, spleen, and blood were isolated and counted from naïve and septic WT mice 6 h after CASP.

In the bone marrow (BM), the number of neutrophils was significantly decreased in septic WT mice (*p* < 0.0001; CASP 22.9 ± 4.5 versus naïve 49.9 ± 5.0) and septic TRAIL–/– mice (*p* < 0.0001; CASP 22.7 ± 4.6 versus naïve 50.0 ± 4.1) (Figure 1a,b). This effect did not occur in the spleen (Figure 1b). In contrast, the number of neutrophils in the blood and in the peritoneal lavage (PL) of septic WT (blood: *p* < 0.01, 79.2 ± 9.1, PL *p* < 0.001, 47.0 ± 16.1) and septic TRAIL–/– mice was highly upregulated (blood: *p* < 0.0001; 79.9 ± 7.4 versus naïve 41.5 ± 12.4, PL: *p* < 0.0001; 49.5 ± 6.7 versus naïve 2.6 ±3.5) (Figure 1b). No difference between WT and TRAIL–/– mice was observed in the spleen.

### 3.2. CASP Surgery Induces a Reactive Left Shift of Neutrophil Granulocytes

Smear slides of blood, BM, and PL were performed from WT mice to analyze morphology changes of neutrophils during polymicrobial sepsis induced by CASP after 1 h, 3 h, 6 h, and 12 h. Representative staining of neutrophils in BM, peripheral blood samples, and PL at 12 h after CASP induction are shown in Figure 2a.

The BM from naïve WT mice presented no segmented but only rod-shaped neutrophils. In CASP mice, the number of segmented neutrophils started to increase at 3 h (*p* < 0.01; 6.3 ± 2.7 versus 0.4 ± 0.1) to 12 h (*p* < 0.001 40 ± 1.6) after CASP. In parallel, the number of rod-shaped neutrophils decreased at 6 and 12 h after CASP treatment, although this effect did not reach statistical significance (13 ± 0.5 and 14 ± 0.4, respectively, versus 17.8 ± 1.3) (Figure 2b). After 1, 3, and 6 h of CASP, the BM contained significantly more rod-shaped than segmented neutrophils (*p* < 0.001 versus segmented). This difference was no longer pronounced after 12 h CASP.

In the blood, naïve WT mice presented no rod-shaped neutrophils. During CASP, the number of rod-shaped (** *p* < 0.01 and *** *p* < 0.001 versus 12 h CASP) and segmented neutrophils (^###^
*p* < 0.001 versus WT CASP 12 h) increased over time (Figure 2c).

In the peritoneal lavage, the number of segmented neutrophils started to increase after 6 h of abdominal sepsis (*p* < 0.05), while the count of rod-shaped neutrophils showed no significant changes (Figure 2d).

### 3.3. Neutrophils of Naïve WT and TRAIL–/– Mice Are Not Sensitive to TRAIL-Stimulated Apoptosis In Vitro

To examine the impact of TRAIL on early and late apoptosis induction, neutrophils isolated from the BM and spleen of naïve WT and TRAIL–/– mice were challenged with different concentrations of recombinant TRAIL in vitro for 1 h. In neutrophils isolated from the BM, TRAIL had no significant impact on early apoptosis. No difference was observed between TRAIL–/– and WT (Figure 3a). In contrast, neutrophils derived from the BM of TRAIL–/– mice showed less late apoptosis and necroptosis compared with the WT (*p* < 0.05) (Figure 3b). In vitro challenging with recombinant TRAIL did not stimulate apoptosis either in TRAIL–/– or in WT neutrophils (Figure 3b). In addition, splenic neutrophils derived from TRAIL–/– mice showed fewer early (*p* < 0.05) (Figure 3c) and late apoptotic or necroptotic cells (*p* < 0.01) compared with the WT (Figure 3d). Again, recombinant TRAIL did not significantly stimulate apoptosis in vitro (Figure 3c,d).

### 3.4. TRAIL-Sensitivity of Neutrophils Is Increased in CASP-Induced Polymicrobial Sepsis

To investigate the impact of TRAIL on the background of sepsis, neutrophils of the BM and spleen were isolated 6 h after CASP induction from WT mice and stimulated with different TRAIL concentrations for 1 h or 12 h in vitro. Neutrophils derived from the BM of septic mice showed a high response to TRAIL in the phase of early apoptosis after 1 h of TRAIL stimulation (*p* < 0.05; TRAIL 100 ng/mL 16.1 ± 3.1%; TRAIL 1000 ng/mL 15.3 ± 2.9% versus TRAIL 0 ng/mL 11.2 ± 2.0%) (Figure 4a). More prolonged stimulation with TRAIL up to 12 h diminished this substantial effect of TRAIL on early apoptosis induction (Figure 4a). The same effect was observed on late apoptosis induction in BM neutrophils (*p* < 0.05; TRAIL 1000 ng/mL 7.3 ± 4.1 % versus TRAIL 0 ng/mL 2.3 ± 0.8%) (Figure 4). Interestingly, when analyzing the effect of TRAIL in neutrophils of the spleen, we found an initial decrease in early apoptotic cells after 1 h TRAIL challenge (*p* < 0.05; TRAIL 1000 ng/mL 7.7 ± 2.3% versus TRAIL 0 ng/mL 14.8 ± 4.8%) (Figure 4c). However, stimulation with 1000 ng/mL TRAIL for 12 h increased the number of early apoptotic neutrophils (*p* < 0.05; TRAIL 100 ng/mL 10.7 ± 2.2%, TRAIL 1000 ng/mL 21.9 ± 3.2% versus TRAIL 0 ng/mL 9.0 ± 1.9%) (Figure 4c). In addition, late apoptosis and necroptosis were significantly increased by 1000 ng/mL TRAIL in neutrophils of the spleen after 1 and 12 h (*p* < 0.05; 1 h TRAIL 1000 ng/mL 25.6 ± 3.3%; 12 h TRAIL 1000 ng/mL 26.6 ± 0.6% versus TRAIL 0 ng/mL 12.9 ± 1.1%) (Figure 4d).

### 3.5. Changes in the Expression of Pro- and Anti-Apoptotic Proteins during Sepsis

To get more insight into the mechanisms of apoptosis inhibition in sepsis, neutrophils of naïve and septic (CASP 6 h) WT and TRAIL–/– mice were isolated, and expression analysis of the pro-apoptotic protein BAX and the anti-apoptotic proteins Bcl-Xl, c-IAP, MCL-1, survivin, and FLIP were performed by Western blotting. Representative Western blots show protein expression levels in neutrophils derived from the BM or spleen (Figure 5a).

In neutrophils of the BM, the expression of the anti-apoptotic protein c-IAP1 was significantly diminished in septic WT mice (*p* < 0.05) and septic TRAIL–/– mice (*p* < 0.05; Figure 5b). In contrast, in neutrophils of the spleen, c-IAP1 seemed to be more expressed in CASP-induced sepsis in WT and TRAIL–/– mice, although this increase did not reach statistical significance (Figure 5b).

CASP also stimulated the expression of MCL-1 in neutrophils derived from the BM and spleen. However, this reached significance only in splenic neutrophils of septic TRAIL–/– animals (*p* < 0.05 versus naïve WT and TRAIL–/–; Figure 5c).

The pro-apoptotic protein BAX was significantly decreased in neutrophils of septic WT and septic TRAIL–/– mice in the BM (*p* < 0.05; Figure 5d). No difference in the expression of BAX was observed in neutrophils of the spleen (Figure 5d).

No significant difference in protein expression was detected for Bcl-xL (Figure 5e). Survivin expression was found only in neutrophils of the BM but not in the spleen (Figure 5a,e). Survivin expression in the BM was not changed by CASP or TRAIL deletion (Figure 5e).

Anti-apoptotic Flip was not expressed in neutrophils derived from the BM of naïve WT and TRAIL–/– mice. However, CASP significantly stimulated the expression of Flip in the BM (*p*< 0.05; Figure 5g). Interestingly, Flip expression was more pronounced in septic WT mice than in septic TRAIL–/– mice (*p* < 0.05; Figure 5g). The same effect was observed in neutrophils derived from the spleen. Here, septic WT mice also showed the strongest Flip expression (*p* < 0.05; Figure 5g).

### 3.6. CASP Induces TRAIL Expression on Neutrophils in the Bone Marrow and Spleen

To analyze the role of TRAIL in CASP-induced sepsis, we compared the expression of TRAIL in neutrophils derived from the BM and spleen of naïve and septic (6 h CASP) WT mice by FACS (Figure 6a). We observed a significant increase of TRAIL-positive neutrophils in septic mice in the BM (*p* < 0.05; 0.3 ± 0.1% versus naïve 0.2 ± 0.1%) and spleen (*p* < 0.05; 1.0 ± 0.5% versus naïve 1.1 ± 0.9%) compared with naïve mice (Figure 6b).

### 3.7. Sepsis Increases the Expression of the TRAIL Receptor DR5 in Neutrophils

TRAIL-receptor DR5, the only TRAIL receptor with a functional death domain in mice, was significantly upregulated in neutrophils of the BM (*p* < 0.01; Figure 7a,b) and spleen (*p* < 0.05; Figure 7a,b) of septic WT mice (6 h CASP) compared with naïve mice. The same effect was observed in TRAIL–/– mice (*p* < 0.05; Figure 7a,b). No difference was detected between septic TRAIL–/– mice and septic WT mice or naïve TRAIL–/– mice and naïve WT mice (Figure 7b). In addition to anti-apoptotic proteins, neutrophils develop resistance to TRAIL by expressing decoy receptors, and we demonstrate organ-specific expression of decoy receptors in sepsis. The DcR2 receptor was significantly upregulated in neutrophils of the spleen of septic WT mice (*p* < 0.05; Figure 7c,d). Although CASP increased DcR2 expression in TRAIL-deficient mice, this failed to show statistical significance (Figure 7d). In neutrophils of the BM, no changes in DcR2 expression were observed (Figure 7d). The DcR1 receptor was slightly upregulated in neutrophils derived from the BM of septic WT mice (*p* < 0.05; Figure 7e,f) but not in TRAIL–/– mice. In the spleen, DcR1 expression did not change (Figure 6f).

## 4. Discussion

The present study elucidates the interaction of TRAIL with the immune system during experimental sepsis (CASP). We observed protein expression regulation of pro-apoptotic Bax and anti-apoptotic MCl-1 and FLIP in septic mice, which might explain the increased apoptosis resistance of these cells leading to increased tissue activity and injury. Our data suggest that this can be antagonized by TRAIL treatment, as we showed that in vitro TRAIL stimulation induced apoptosis in neutrophil granulocytes derived from the spleen and, to a smaller extent, in neutrophils derived from the bone marrow of septic mice. This was associated with an increased upregulation of TRAIL receptor expression in these cells. In contrast, neutrophils of non-septic mice were resistant to TRAIL-induced apoptosis and showed no receptor upregulation. These findings suggest that TRAIL-induced apoptosis regulates neutrophil survival during peritonitis.

First, we showed that CASP treatment reduces the number of neutrophils in the bone marrow, which points to neutrophil mobilization and migration. While the number of neutrophils in the spleen did not change significantly in CASP-induced sepsis, we showed highly elevated numbers of neutrophils in the blood and the peritoneum as the site of the infection. Using the morphology of neutrophils, we examined the development status of the cells. Rod-nucleated neutrophils are an early form of neutrophils, and increased numbers of these in the blood, known as left shift, indicate a response to infection. Segment-nucleated neutrophils, on the other hand, as the last stage of maturation of granulopoiesis, increase during the timeline of CASP as a consequence of the increased release of immune cells. As expected, we found a left shift of neutrophils in the blood as well as leukocytosis in the blood and the peritoneal lavage during CASP-induced peritonitis. This is a clear sign of inflammation and proves sepsis induction in our mouse model. The decreasing ratio of rod-shaped neutrophils in the bone marrow designates the bone marrow as the origin of a large reservoir of neutrophils. Maturation and subsequent migration of neutrophils might explain their decreasing frequency in the bone marrow during sepsis. The increased number of segmented neutrophils indicates reactive myelopoiesis.

In the next step, we investigated the impact of TRAIL on apoptosis induction in neutrophils isolated from the bone marrow and the spleen. In vitro, TRAIL does not impact apoptosis induction in neutrophils isolated from the bone marrow and the spleen of naïve WT and TRAIL–/– mice, which is an important aspect regarding the therapeutic application of TRAIL.

However, in septic mice, in vitro stimulation with TRAIL induces apoptosis in spleen and bone marrow neutrophils. This finding supports the hypothesis that the improved survival of TRAIL-treated mice in abdominal sepsis [6] is influenced by the interaction of neutrophil granulocytes with TRAIL. These data are supported by Yoo et al., who reported that the plasma TRAIL level and the severity of sepsis are inversely correlated [13]. However, in murine peritonitis, the number of neutrophilic granulocytes in the peritoneum, as the region of the infection, was increased by TRAIL-treatment compared with placebo treatment [14]. These observations led to the hypothesis of an increasing TRAIL sensitivity of neutrophils in sepsis. Interestingly, this might be time- and concentration-dependent, as we observed for the pleiotropic effects of TRAIL in our study. In the early stages of peritonitis, neutrophilic granulocytes of the spleen showed an increased anti-apoptotic reaction after TRAIL treatment, which was reversed with a longer duration of peritonitis. The rate of late apoptosis increased in neutrophils of the spleen with a higher concentration of TRAIL. Interestingly, this effect was much less pronounced in the bone marrow neutrophils of septic animals, where higher doses of TRAIL and longer stimulation durations were required. These findings support the hypothesis of the pleiotropic effects of TRAIL [15].

In this context, we demonstrated the protective role of TRAIL-induced apoptosis in the early stage of abdominal sepsis. These data are supported by the findings that TRAIL is expressed in resting neutrophils, and its expression is enhanced at the mRNA and protein level after stimulation with IFN-alpha and gamma [16]. In contrast, TRAIL does not induce a chemotactic response in neutrophils [17]. Steinwede et al. demonstrated that apoptosis of lung macrophages in pneumococcal pneumonia is mediated by neutrophil-derived TRAIL in mice [18]. TRAIL’s signaling comprises a complex pathway with at least six known cascades converging and diverging, presumably contributing to an overall intracellular pro- or anti-apoptotic milieu [19].

Within the present study, we showed that the TRAIL-receptor Dr5, the only one with a functional death domain, was strongly upregulated in neutrophils of the spleen and bone marrow after 6h CASP treatment of WT and TRAIL–/– mice. Furthermore, TRAIL expression was significantly increased in neutrophilic granulocytes of septic WT mice.

To understand the mechanisms of apoptosis inhibition in peritoneal sepsis, we examined the expression level of pro- and anti-apoptotic proteins. The pro-apoptotic protein BAX was diminished under CASP treatment in the bone marrow, while the anti-apoptotic proteins MCL-1 and Flip were upregulated. The expression of MCL-1 and Flip was also increased in splenic neutrophils of septic mice. Mcl-1 is known as an antagonist of BAX [20]. Paunel-Gorgülu et al. reported that neutrophils from trauma patients expressing high levels of intracellular Mcl-1 are resistant to apoptosis in response to the pro-apoptotic stimulus staurosporine [21]. Our findings of strongly elevated intracellular Mcl-1 levels in neutrophilic granulocytes in the abdominal sepsis and the upregulated apoptosis rate of septic neutrophils after TRAIL treatment highlight the hypothesis of Dr5-mediated apoptosis. C-FLIP causes resistance to pro-cell death signals [22] and probably promotes apoptosis resistance to the neutrophils in the early stage of sepsis. These findings support the hypothesis of the TRAIL-receptor DR5-mediated apoptosis. These findings might explain the prolonged survival of neutrophils in sepsis.

Regarding the higher expression of DR-5 and the anti-apoptotic proteins, we found no difference between WT and TRAIL–/– mice. We speculate that the elevated apoptosis rate of neutrophils was caused by exogenous TRAIL treatment with no influence from endogenous TRAIL levels. Hence, TRAIL–/– mice did not show increased apoptosis in naïve mice after TRAIL treatment compared with septic WT mice. This implies that the effects of improved survival rate in sepsis are induced by exogenous TRAIL and not endogenous TRAIL. The finding that TRAIL deficiency did not alter the number of infiltrating neutrophils but significantly decreased the number of apoptotic neutrophils during sepsis supports this theory [23].

McGrath et al. showed an increased proportion of apoptotic neutrophils detected in BAL following TRAIL treatment in a model of acute pulmonary inflammation [24], which supports the therapeutic effect of TRAIL in sepsis, peritonitis, and pulmonary inflammation.

TRAIL is known to exert pluripotent effects in sepsis, leading to an improved survival rate in the early stage of abdominal sepsis and a decreased survival rate in the late phase of hypoinflammation [25]. Therefore, the therapeutic approach of blocking anti-TRAIL in septic mice improves the control of secondary bacterial infection and restores CD8 T cell responses [26].

In the present study, we showed that the protective effect of TRAIL in the colon ascendens stent peritonitis [23] is caused by apoptosis induction in neutrophil granulocytes. This might explain our former observation of reduced tissue injury in TRAIL-treated septic mice [8] and supports the finding that depletion of neutrophils decreases sepsis survival and abolishes therapeutic TRAIL effects during CASP [8].

## 5. Conclusions

In this study, we characterized the fundamental features in the discussion of the therapeutic benefits of TRAIL treatment in the early stage of abdominal sepsis by the interaction with neutrophil granulocytes. Our data paves the way for a deeper understanding of the contribution of neutrophils to inflammatory conditions.

In conclusion, the present study shows that neutrophilic granulocytes display an increased sensitivity to TRAIL-induced apoptosis in the murine CASP model, which is associated with an increased expression of the TRAIL receptor DR5. These findings could explain the improved survival of TRAIL-treated mice during sepsis [6] and support further investigations with respect to a therapeutical application of TRAIL in abdominal sepsis.

## Figures and Tables

**Figure 1 cancers-15-01773-f001:**
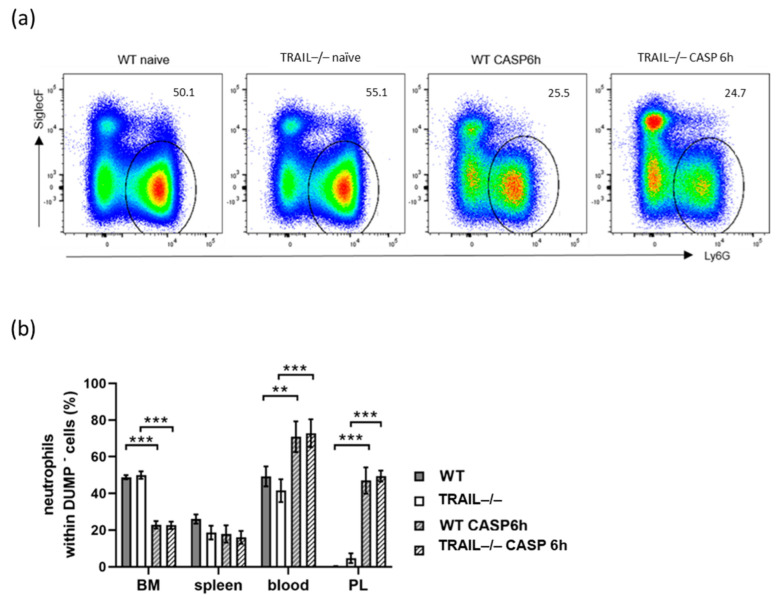
Influence of 6h CASP on the neutrophil ratio in lineage-negative Ly6G cells. (**a**) Representative FACS plots show staining of neutrophils derived from the bone marrow. (**b**) The numbers of neutrophils within DUMP^–^ cells of 5 mice each are shown for bone marrow (BM), spleen, peritoneal lavage (PL), and blood. Neutrophils of septic WT and TRAIL–/– mice were compared with naïve ones. Graphs show mean ± SEM (** *p* < 0.01, *** *p* < 0.001).

**Figure 2 cancers-15-01773-f002:**
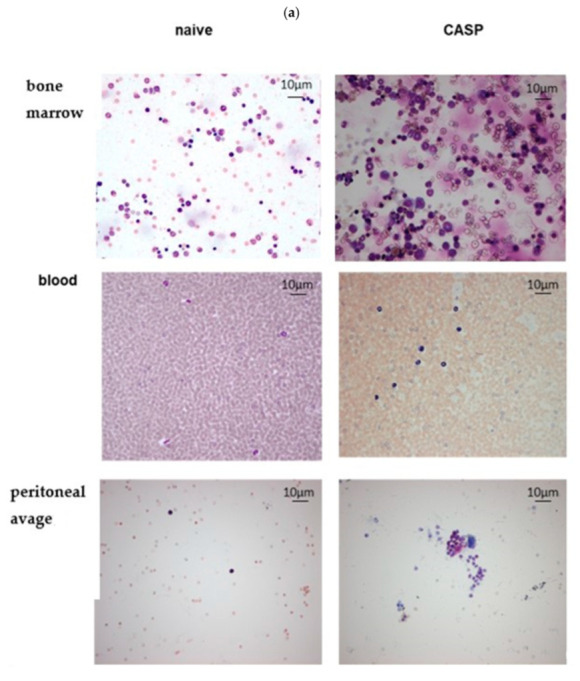
CASP induces a reactive left shift of neutrophil granulocytes. Neutrophil granulocytes of untreated WT and septic WT mice 1 h, 3 h, 6 h, and 12 h after CASP induction were counted in high-power fields (HPF) of smear slides (*n* = 5). (**a**) Representative smear slides of naïve WT and septic WT after 12 h CASP stained with Hemacolor (original magnification 400×). (**b**) Maturation of neutrophil granulocytes during CASP in BM (*** *p* < 0.001), (**c**) blood (* *p* < 0.01 and ** *p* < 0.001 versus WT CASP 12 h rod-shaped; ^###^
*p* < 0.001 versus WT CASP 12 h segmented), and (d) peritoneal lavage (* *p* < 0.05 and ** *p* < 0.01 versus WT CASP 12 h segmented). Graphs show mean ± SEM.

**Figure 3 cancers-15-01773-f003:**
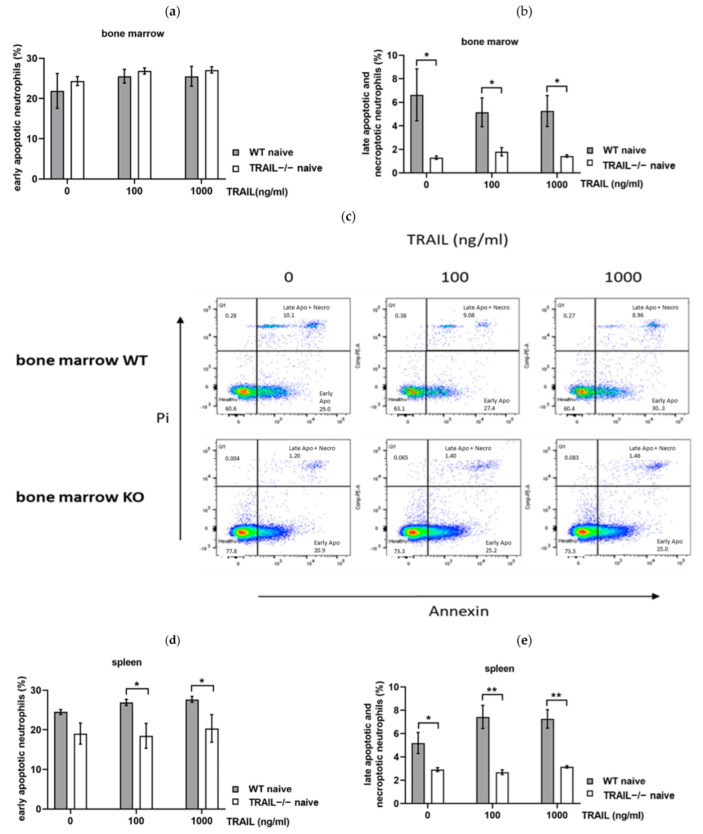
Recombinant TRAIL did not stimulate apoptosis in neutrophils derived from naïve mice in vitro. Neutrophils isolated from bone marrow (**a**,**b**) and spleen (**d**,**e**) of naïve WT and TRAIL–/– mice (*n* = 5 each) were stimulated with different concentrations of TRAIL (0 ng/mL, 100 ng/mL, 1000 ng/mL) in vitro. Representative FACS plots show staining of neutrophils derived from the bone marrow (**c**) and the spleen (**f**). The level of early (**a**,**c**) and late (**b**,**d**) apoptosis in neutrophils with and without TRAIL treatment are shown. Graphs show means ± SEM (* *p* < 0.05, ** *p* < 0.01).

**Figure 4 cancers-15-01773-f004:**
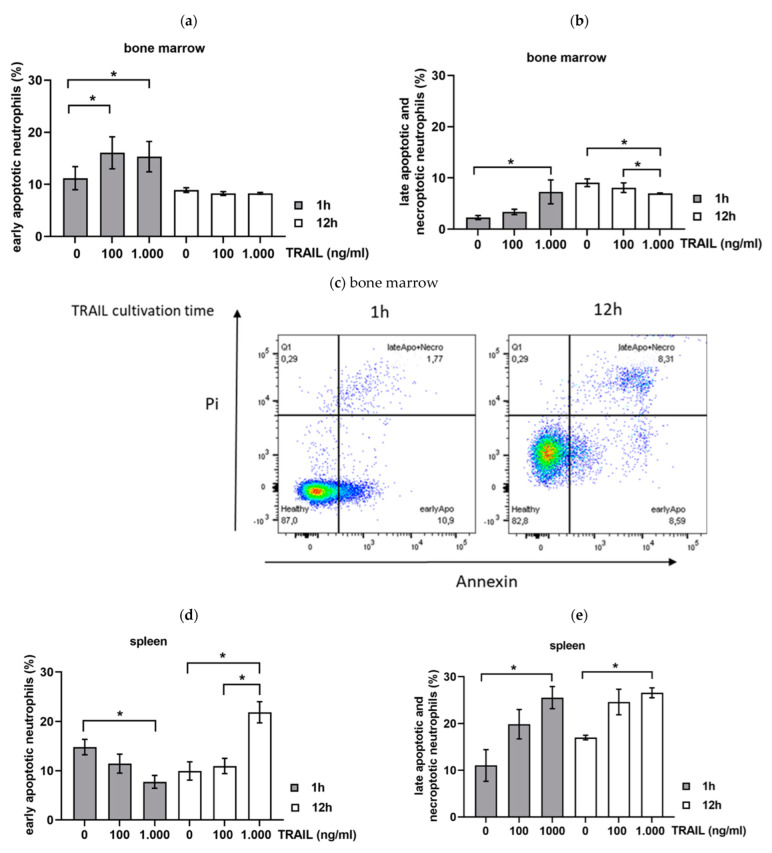
Impact of TRAIL stimulation on apoptosis in neutrophils derived from BM and spleen of septic WT mice. Neutrophils of septic WT mice 6 h after CASP induction (*n* = 5) were cultured for 1 and 12 h with and without recombinant TRAIL in vitro. (**a**) Percentage of early apoptotic and (**b**) late apoptotic and necroptotic neutrophils of BM, (**d**) early apoptotic and (**e**) late apoptotic and necroptotic neutrophils of the spleen are shown (*n* = 5 each). Representative FACS plots show staining of neutrophils derived from the bone marrow (**c**) and the spleen (**f**). Graphs show mean ± SEM (* *p* < 0.05).

**Figure 5 cancers-15-01773-f005:**
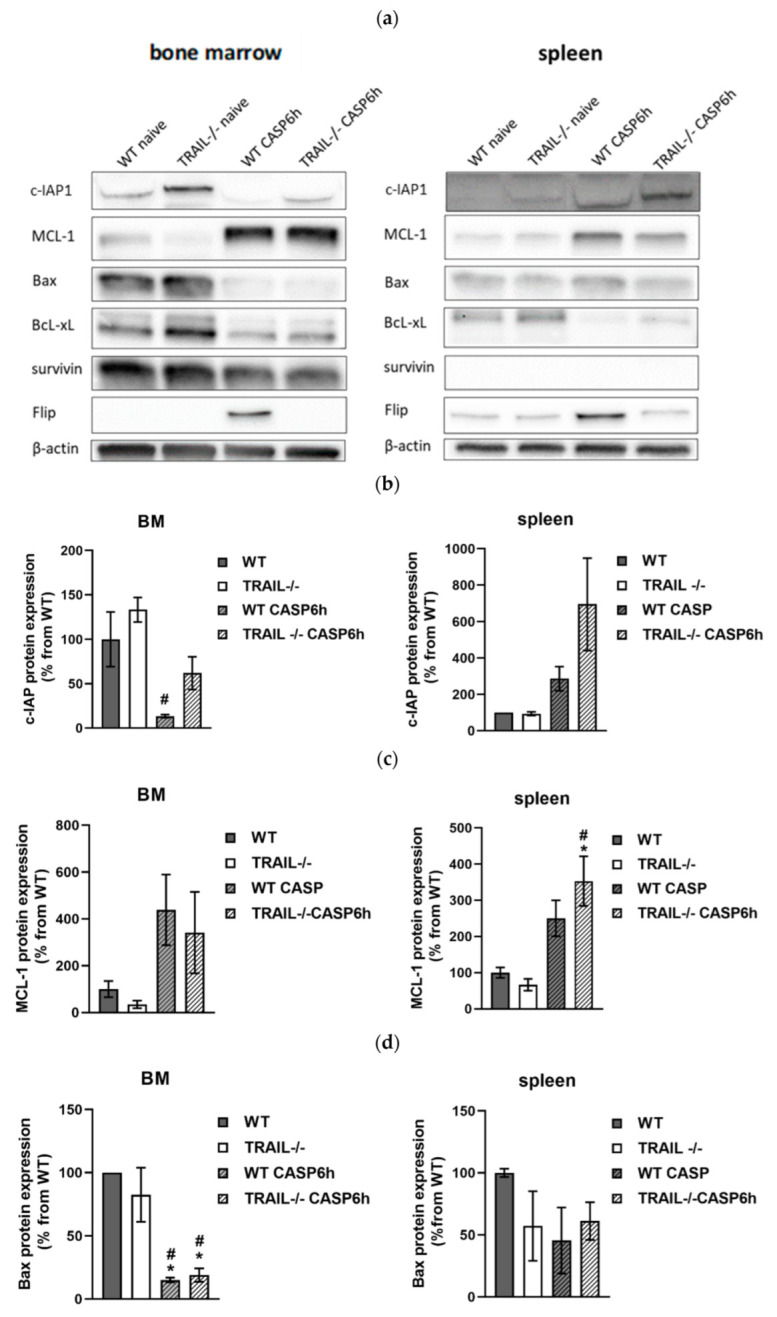
Effect of 6h CASP on the expression of pro- and anti-apoptotic proteins. (**a**) Representative Western blots show protein expression in neutrophils derived from the BM and spleen of naïve and septic WT or TRAIL–/–mice. Quantification of the expression level of c-IAP (**b**), MCL-1 (**c**), Bax (**d**), BcL-xL (**e**), survivin (**f**), and Flip (**g**). Graphs show mean ± SEM (* *p* < 0.05; versus WT ^#^
*p* < 0.05 versus TRAIL–/–).

**Figure 6 cancers-15-01773-f006:**
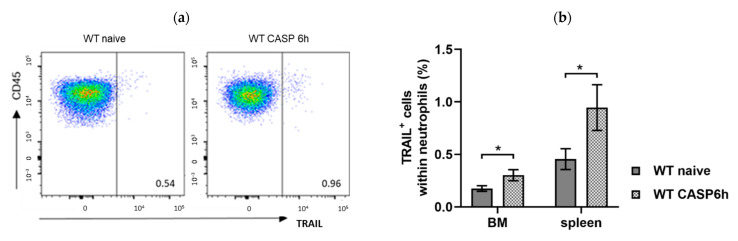
Impact of 6h CASP on the expression of TRAIL in neutrophils from BM and spleen. TRAIL was stained and assessed by FACS. (**a**) Representative FACS plots show TRAIL-positive neutrophils. (**b**) Quantification of TRAIL expression in neutrophils of naïve and septic WT mice (*n* = 5 each). Graphs show mean ± SEM (* *p* < 0.05).

**Figure 7 cancers-15-01773-f007:**
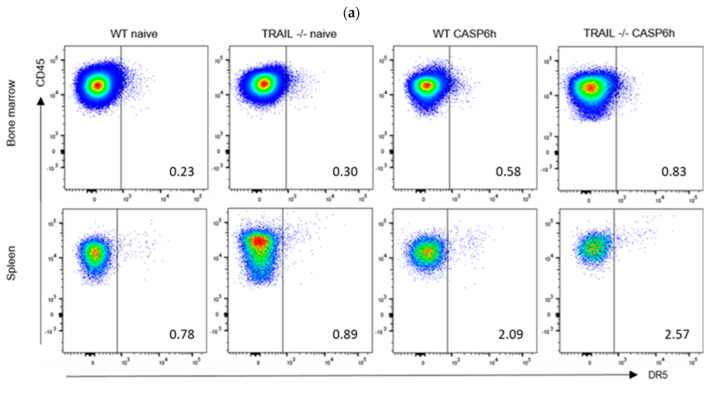
Effect of 6h CASP treatment on the expression of TRAIL receptors. TRAIL receptors DR5, DcR1, and DcR2 were stained and assessed by FACS. Representative FACS plots show (**a**) DR5 staining in neutrophils derived from BM and spleen, (**c**) DcR2 staining in neutrophils of the spleen, and (**e**) DcR1 staining in neutrophils of the BM. Quantification of (**b**) DR5, (**d**) DcR2, and (**f**) DcR1 expression in neutrophils of the BM and spleen of *n* = 5 animals in each group. Neutrophils of septic WT and TRAIL–/– mice were compared with naïve ones. Graphs show mean ± SEM (* *p* < 0.05, ** *p* < 0.01).

## Data Availability

A dataset of the present study can be requested from the corresponding author on reasonable request.

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
