# Peer review of "The Impact of TRAIL on the Immunological Milieu during the Early Stage of Abdominal Sepsis"

_cancers, 2023, doi:10.3390/cancers15061773_

Round 1

Reviewer 1 Report

Tumor necrosis factor-related apoptosis-inducing ligand (TRAIL) plays a role in inducing apoptosis in certain cells. The ability of TRAIL as a potential therapeutic target due to its ability to selectively kill damaged cells was studied with mixed outcomes and conclusions.

The current study has shown that TRAIL did not have an apoptotic effect on healthy isolated neutrophils but showed an apoptotic effect on neutrophils of the spleen and bone marrow of scepsis mice suggesting the role of TRAIL and its interaction with neutrophil granulocytes in the survival of mice with abdominal sepsis.

Overall, this paper is a significant contribution to understanding the role of TRAIL in sepsis. 

Some minor language editing might be necessary to make the flow of text better.

Reviewer 2 Report

I went through the manuscript entitled as “The impact of TRAIL during the early stage of abdominal sep-2 sis on the immunological milieu” by Dr Byer and co-workers. Authors investigated that Based on the comparison of the neutrophil isolated from BM, spleen, periphery, and peritoneum with or without Trail gene, neutrophil will have different degree of TRAIL dependency for the cell death. The topic is important in both basic and clinical point of view, and the study, in general, appear to be carefully designed and performed carefully.

Partly due to the complexity of the study design, and the differences are relatively small ( as low as few % in some of the comparison). Therefore, comparison of the neutrophil population from BM, spleen, peritoneum in both WT and Trail -/- genotypes and with/without CASP, it is very difficult to follow the results/logic/interpretation. Since most of the results appear to be the comparison between each group are with very small differences in different cell population at different time point, the referee feels that it may be half full and half empty type of discussion so that there seems several different ways of interpretation from the data presented depending on the comparison to focus on.

The referee feel that with this small differences of each groups, if authors analyzed the same samples with different time point with different ways of groupwise classifications, the interpretation could be somewhat different. Early apoptosis and late apoptosis, authors are suggested to show the original FACS instead of showing the number of the population of each groups. It is difficult to understand the trend of induction of cell death in each groups. 

Authors obtained some certain ways of results differences of different cellular populations. Authors showed the comparison of the results from the Early apoptosis and late apoptosis, as there is very small fraction of differences between these comparison, the result would be different if authors analyzed different time point, as the cell death pathways is moving on with the time course.

The results and the overall conclusion, in general, appear to be somewhat predictable, in my view. As these are the situation, how this observation would contribute back to the clinic is uncertain.

Authors are encouraged to make better organized the figures/panels based on the logic with clearer and more focused the comparison. Some figure, for example including Fig2/3/4 many panels are just listed up which may not necessarily be associated with the mainstream logic authors are intent to discuss. The figures may better to present the results more focused, and organized manner for the comparison, with more focused direction of interpretation along with the potential varieties of possible interpretation.  

Fig 1b; It is not clear the term “ frequency” means in this bar graph. The % presented is relative to what ? What is the main population of the comparison?

Fig 2a-d; It would be easy to follow if authors put “WT” on the top of the figure. Similar to other figures as well. For example, put BM on top of Fig 2b, Blood on top of the panel. What is blood population mean here is not clear. Does “blood” mean neutrophil from the peripheral blood samples? Then what is “frequency” mean?

Fig 2a: bone marrow; it seems “bonemarro” missing “w”.Put WT on the top of the panel will be easy to catch up the results for the reader.

Fig 3.  I would suggest authors to present the original FACS panel as well. The interpretation of this panel would be complex, and the referee feel that there are many numbers of ways for the interpretation will be possible, depending on what authors see the results as the observation appear to be relative comparison.

Fig 4. For example, the result presented could be different if authors harvested the neutrophil in different time point other than 6 hours.

Fig 5. Fig 5f, Y axis could be (% from WT? why control only this panel?)

 The referee feel that if harvest the cells at different time point from the same cell population, the results would be different. There are many numbers of different ways of interpretation would be possible from the results presented, it seems like.

Fig 6-7. The figures may better to present the results more focused, and organized manner for the comparison, with more focused direction of interpretation along with the potential varieties of possible interpretation. As there is very small fraction of differences between these comparison, the result would be different if authors analyzed different time point, as the cell death pathways is moving on with the time course.

Round 2

Reviewer 2 Report

The revised manuscript appear to address most of my previous concerns with satisfactory manner. The topic is important in both basic and clinical point of view, and the study and the presentation of the revised manuscript, in general, appear to be much more improved for the general readership of Cancers.